# Demystifying Singular Defects in Large Language Models

**Haoqi Wang** [1]   **Tong Zhang** [1,2]   **Mathieu Salzmann** [1,3]

## Abstract

Large transformer models are known to produce high-norm tokens. In vision transformers (ViTs), such tokens have been mathematically modeled through the singular vectors of the linear approximations of layers. However, in large language models (LLMs), the underlying causes of high-norm tokens remain largely unexplored, and their different properties from those of ViTs require a new analysis framework. In this paper, we provide both theoretical insights and empirical validation across a range of recent models, leading to the following observations: i) The layer-wise singular direction predicts the abrupt explosion of token norms in LLMs. ii) The negative eigenvalues of a layer explain its sudden decay. iii) The computational pathways leading to high-norm tokens differ between initial and noninitial tokens. iv) High-norm tokens are triggered by the right leading singular vector of the matrix approximating the corresponding modules. We showcase two practical applications of these findings: the improvement of quantization schemes and the design of LLM signatures. Our findings not only advance the understanding of singular defects in LLMs but also open new avenues for their application. We expect that this work will stimulate further research into the internal mechanisms of LLMs. Code is released at https://github.com/haoqiwang/singular_defect.

## 1. Introduction

While large foundation models are proving increasingly effective, their intrinsic behavior remains poorly understood. An example of this is the emergence of tokens whose norm is unexpectedly higher than that of the majority of the other tokens during the forward pass, as illustrated in Fig. 1. This behavior has been observed in both large vision transformers (Darcet et al., 2024; Wang et al., 2024) and large language models (LLMs) (Sun et al., 2024).

In particular, in vision foundation model DINOv2 (Oquab et al., 2024; Darcet et al., 2024), high-norm tokens appear as defective patch tokens, degrading the feature quality and hindering performance on downstream dense prediction tasks. In pursuit of repairing these defects, Wang et al. (2024) revealed a connection between the direction of the high-norm tokens and the leading left singular vector of the matrix that provides a linear approximation of a transformer layer, and therefore name these tokens as *singular defects*.

In the context of LLMs, high-norm hidden states have also been observed (Sun et al., 2024) and referred to as *massive activations*. Specifically, the following properties were empirically identified: (1) The appearance of a massive activation is abrupt, emerging suddenly in one layer and diminishing later in the model after another. (2) Massive activations appear mostly for the initial token and the delimiter tokens. These properties differ from those made with DINOv2, where the norm of the defective tokens increases gradually layer by layer, and the defective tokens are randomly scattered across the feature map, with a tendency to appear in low-semantic regions. Given these different behaviors of high-norm tokens in LLMs *vs*. ViTs, it is natural to wonder *(i) whether the theory of singular defects can be applied to LLMs; and (ii) how to further explain the new observations related to massive activations?*

In this paper, we confirm that singular defects can predict the direction of high-norm tokens in LLMs. However, it falls short of explaining the high-norm properties that are unique to LLMs. To understand the full life cycle of the high-norm tokens, we thus expand the theory and provide the following insights:

1. (*Development*) The explosion of the initial[1] token norm is linked to self-attention, whereas that of the noninitial high-norm tokens is unrelated to self-attention.

2. (*Trigger*) A norm explosion is initiated when the input vector has a projection onto the leading right singular

---

[1]We define the initial token as the first token in the user input.

[1]School of Computer and Communication Sciences, EPFL, Switzerland [2]University of Chinese Academy of Sciences, China [3]Swiss Data Science Center, Switzerland. Correspondence to: Mathieu Salzmann <mathieu.salzmann@epfl.ch>.

*Proceedings of the 42$^{nd}$ International Conference on Machine Learning*, Vancouver, Canada. PMLR 267, 2025. Copyright 2025 by the author(s).

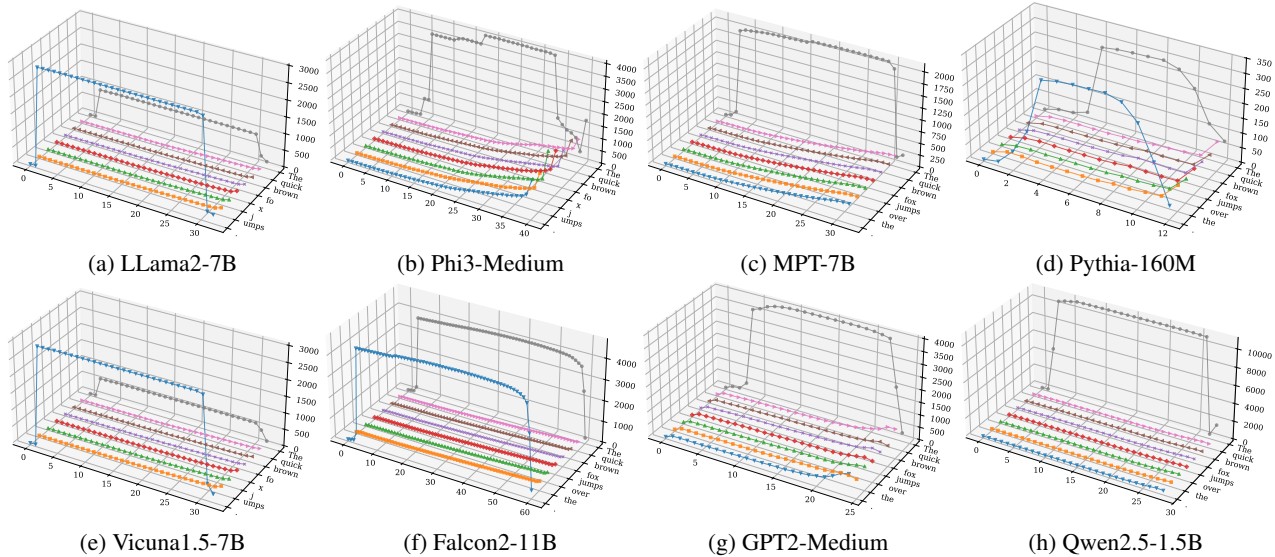

*Figure 1.* High norm tokens in various LLMs. Each subfigure plots the norm of the first few tokens and the token '`.`' in the sentence '`The quick brown fox jumps over the lazy dog.`'. The $x$-axis is the layer id, the $y$-axis shows different tokens, and the $z$-axis is the norm. Layer 0 is the input embedding layer, and the others are transformer layers. Additional models are shown in Appendix A.

vector of the linear approximation of the explosion layer's feed-forward network (FFN) module.

3. (*Explosion*) Once triggered, the high-norm token in the layer's output aligns with the direction of the layer-wise singular defect direction.

4. (*Decay*) The layer that decays the high-norm tokens has a negative eigenvalue associated with an eigenvector that aligns with the singular defect direction.

We empirically validate these findings on a variety of LLMs, including LLaMA2 (Touvron et al., 2023), Phi3 (Abdin et al., 2024), MPT (Team, 2023), Pythia (Biderman et al., 2023), Vicuna1.5 (Platzer & Puschner, 2021), Falcon2 (Malartic et al., 2024), GPT2 (Radford et al., 2019), Qwen2.5 (Team, 2024), to name a few.

For the behavior of high-norm tokens during training, our experiments reveal that the direction of high-norm tokens gradually stabilizes in training and remains consistent even after fine-tuning. Moreover, we conjecture that the causal self-attention mechanism is one of the defining factors for the emergence of high-norm tokens.

Finally, we demonstrate that a better understanding of the high-norm tokens in LLMs can lead to novel applications. *i) High-Norm Aware Quantization Design*. Outlier activations induced by high-norm tokens cause significant performance in low-bit quantization. To mitigate this, we propose a high-norm aware quantization strategy that selectively preserves precision for these critical layers, improving robustness without compromising efficiency. *ii) LLM Signature*

*via Singular Defects*. The singular defect direction, which stabilizes in the late training stages and persists through fine-tuning, serves as a robust model signature. This signature enables distinguishing whether an LLM was fine-tuned from another model and thus detect model infringement.

Ultimately, we believe that understanding singular defects will not only stimulate novel applications but also spur new insights into the internal mechanism of LLMs.

## 2. Related Work

**Large Language Models (LLMs).** Transformer-based LLMs (Minaee et al., 2024) have achieved remarkable performance in various natural language processing tasks. They are mainly pre-trained with causal language modeling (*e.g.*, LLaMA2 (Touvron et al., 2023)) or masked language modeling (*e.g.*, BERT (Devlin et al., 2019)). BERT-like models employ a bidirectional attention mechanism to learn contextual representations, typically adopting an encoder-only architecture. In contrast, LLaMA-like models utilize a decoder-only architecture and are trained autoregressively with causal self-attention, modeling the probability distribution of the next token in a sequence. In our observations, LLaMA-like models exhibit extremely high-norm tokens in their hidden states, whereas BERT-like models do not. In this work, we focus on LLaMA-like models and investigate the characteristics of these high-norm tokens.

**High-Norm Tokens in ViTs and LLMs.** Several recent works have observed the presence of high-norm defective tokens in the feature maps of ViTs (Darcet et al., 2024;

Wang et al., 2024) and proposed methods to repair them. In particular, Wang et al. (2024) used the leading left singular vector of the matrix representing a linear approximation of the transformer layer to predict the direction of the defective tokens. In the context of LLMs, Sun et al. (2024) noticed that certain activations in the hidden states have a huge magnitude (*i.e.*, massive activations). They observed that massive activations are consistently present in very few fixed dimensions but did not provide a mathematical explanation. To account for the different properties of high-norm tokens in ViTs and LLMs, we extend the theory of Wang et al. (2024) to LLMs, providing a systematic explanation for their high-norm tokens. Furthermore, while Sun et al. (2024) study massive activations from the perspective of the individual scalar values within a token, we analyze high-norm tokens as a whole vector and provide a mathematical framework to explain high-norm tokens in LLMs.

**Applications of High-Norm Tokens.** While recent studies have identified high-norm tokens in LLMs, their practical implications remain largely unexplored. We highlight two key applications: Quantization and Model Signature. High-norm tokens induce activation outliers, posing challenges for tensor-wise quantization methods (Dettmers et al., 2022; Xiao et al., 2023; Grattafiori et al., 2024). Additionally, as more open-source LLMs become available, tracing model lineage is increasingly important. Prior works (Yu & Wang, 2024; Mu et al., 2024) investigate model tracing through internal representations, yet, they can only deal with models in vision. To the best of our knowledge, we are the first to leverage insights from the high-norm token analysis to address both challenges in LLMs.

## 3. Analysis of High-Norm Tokens in LLM

As shown in Fig. 1, most recent LLMs manifest high-norm tokens in intermediate layers, regardless of the model size, training data, variations in model architecture, etc. Echoing the observations in (Sun et al., 2024), we note that the high-norm tokens appear abruptly in a certain layer and decay suddenly after another layer, with the initial token consistently having a high norm in the middle layers, while certain later tokens, such as the delimiter '.', also exhibit a high norm in some models. These phenomena differ from those in vision transformers, where norms gradually increase without sudden drop (see Figure 4 in (Darcet et al., 2024)), and high-norm tokens appear at random spatial locations.

This section provides a comprehensive understanding of the distinctive high-norm phenomenon in LLMs by expanding the theory of singular defects that was originally developed for ViTs. As illustrated in Fig. 2, we study this phenomenon from four perspectives, covering the full life cycle of the high-norm tokens: (1) Development: the computational

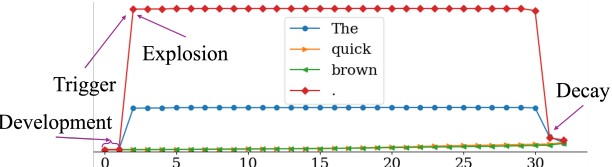

*Figure 2.* The four properties of high-norm tokens. The $x$-axis is the layer id, and the $y$-axis is the norm of the tokens. The results are obtained from LLaMA2-7B.

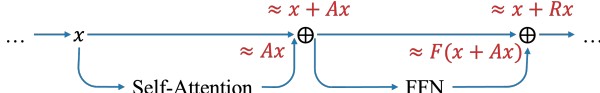

*Figure 3.* Transformer layer in LLaMA2. Given an input token $x$, we show the approximate output of each module in red.

pathways leading to norm increases. (2) Trigger: the cause of the norm increase just before the explosion layer. (3) Explosion: the appearance of high-norm tokens in intermediate layers and their correlation with network parameters. (4) Decay: how high-norm tokens disappear.

### 3.1. *Explosion*: Appearance of High-Norm Tokens

The most obvious pattern in Fig. 1 is that the hidden states of some tokens have extremely high norms in the intermediate layers, and so consistently across different LLMs. Interestingly, we find that for each model, all the high-norm tokens share *the same direction*, regardless of the input text, the layer, and the position of the token in the sentence. We define the average of the high-norm tokens as the *empirical high-norm direction*.

Taking LLaMA2-7B as an example, we extract the hidden states of 1K random rows from the `WikiText2-v1` dataset (Merity et al., 2017) across all layers and compute the norm of each token in each layer. We collect all hidden states with a norm larger than 500. The average pairwise angle between any two high-norm tokens is only 3.1 degrees, which verifies our claim. Additional statistics for more models are provided in Appendix C, demonstrating that this phenomenon is general across models. Sun et al. (2024) observed that the massive activations appear at fixed feature dimensions, which echos our observation from the perspective of vector direction.

Notably, the behavior that the same high-norm direction appears across layers differs from DINOv2 where the direction of the high-norm tokens varies across layers. Despite the distinctions between high-norm tokens in ViTs and LLMs, we next verify whether the theory of singular defects developed for DINOv2 is applicable to LLMs.

Let us first review the basic concepts introduced in (Wang et al., 2024). The core of the theory of singular defects is

the linear approximation of transformer layers under the single-token assumption. It assumes that only a single token is provided as input, so that the interaction across tokens in self-attention can be ignored, making the mathematical analysis tractable. Note that most recent LLMs are trained with causal self-attention, where a token can only attend to itself and the previous tokens. Thus, the inference of the first token perfectly matches the single-token assumption.

We use LLaMA2 as an example to illustrate how to approximate the transformer layer as a linear operator. The structure of the transformer layer is shown in Fig. 3, where the self-attention module and the FFN module are the two residual components. The self-attention module can be approximated as a matrix-vector multiplication,

$$\text{Attention}(x) \approx Ax := A_2 A_1 A_0 x, \quad (1)$$

where $A_0$ is a diagonal matrix representing the weight of the attention RMSNorm, $A_1$ is the weight matrix of the value projection, and $A_2$ is the weight matrix of the output projection. Note that for LLMs that use a different self-attention design, the approximation of the self-attention module can be adjusted accordingly. For example, LLaMA3 uses a grouped query attention (Ainslie et al., 2023) whose key-value heads are repeated. Then, we can, accordingly, repeat the weight matrix of the value in the approximation.

The FFN module can be approximated as

$$\text{FFN}(x) \approx Fx := F_2 F_1 F_0 x, \quad (2)$$

where $F_0$ is a diagonal matrix representing the weight of the feed-forward RMSNorm, $F_2$ is the weight matrix of the down_proj, and $F_1$ is the least-square linear approximation (see (Wang et al., 2024) for details) of the nonlinear function $F_1 x \approx \text{silu}(W_1 x) \odot (W_3 x)$, in which $W_1$ is the gate_proj and $W_3$ is the up_proj. For LLMs that use alternative FFN designs, e.g., $F_1 x \approx \text{GELU}(W_1 x)$ in Pythia, the least-square linear approximation of the FFN module can be adjusted accordingly.

Combining the two modules with the identity paths, the transformer layer can be approximated as

$$\text{Layer}(x) \approx Lx := x + (A + F + FA)x =: (I + R)x, \quad (3)$$

where the right-hand side decomposes the approximated matrix into an identity path $I$ and a residual path $R$. The layer-wise singular defect direction is then defined as the leading left singular vector of the matrix $L$ for each layer.

We compute the layer-wise singular defect directions of LLaMA2-7B and compare them with the empirical high-norm direction. The acute angles between the predicted directions and the empirical one are provided in Fig. 4. We observe layer-2 and layer-31 to yield very small angles (6.05 and 2.93 degrees, respectively) with the empirical

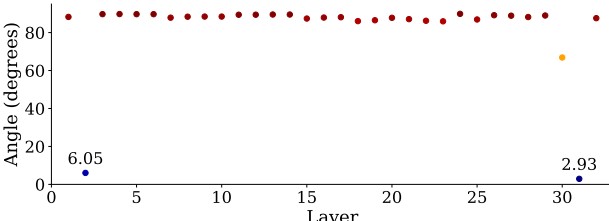

Figure 4. Acute angle between the predicted layer-wise singular defect directions and the empirical high-norm direction for LLaMA2-7B. The predictions for more LLMs are provided in Appendix D.

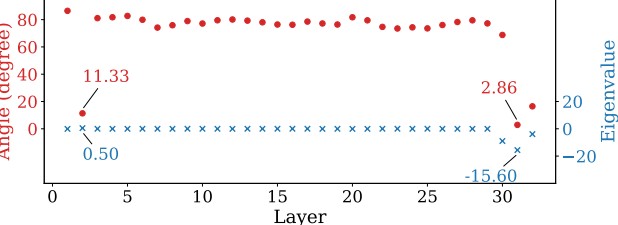

Figure 5. For LLama2-7B, the minimum angles between the eigenvectors of $R$ and the empirical high-norm direction are shown in red, and the corresponding eigenvalues are shown in blue. More examples are shown in Appendix E.

high-norm direction. These two layers correspond to the layer that increases the norm and decreases the norm of the tokens, respectively (see Fig. 1a). Layers 3–30 only cause small perturbations to the high-norm tokens. Thus, the high-norm tokens in the hidden states are created by layer-2, then preserved by the identity path between layer-3 and layer-30, and suppressed by layer-31. This explains why the set of high-activation channels observed in (Sun et al., 2024) is *fixed*: the high-norm tokens are (nearly) unchanged in the intermediate layers. Unsurprisingly, we observe that the angles between the layer-wise predicted directions and the empirical directions are large, as they do not modify the high-norm tokens. Note that the accumulated singular defect directions (see (Wang et al., 2024) for a formal definition) may mislead us about the contribution of the intermediate layers to the high norms. Therefore, in the remainder of the paper, we will focus on layer-wise singular defect directions.

### 3.2. *Decay*: Eigenvalues of Decay Layers

The leading left singular vector of the matrix $L$ encodes the output direction having the largest norm for all possible unit-length inputs, and the corresponding singular value is the norm of the output in that direction. However, this interpretation contrasts with the behavior of the *decay layer*, which greatly reduces the norms of high-norm tokens. To better describe the behavior of such a layer, we instead propose to use eigenvalue and eigenvector decomposition.

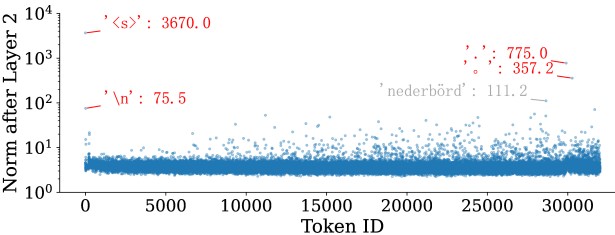

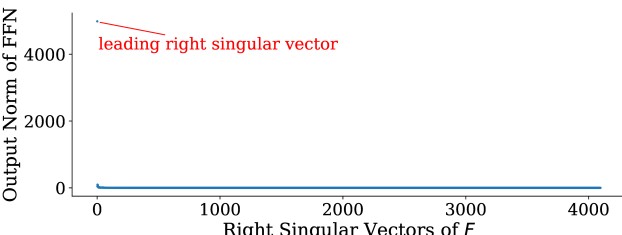

*Figure 6.* Attention-independent exploding path of LLaMA2-7B. The $y$-axis is the norm of each token at the output of layer-2 after removing all self-attention blocks from the model. The largest five tokens together with their norms are annotated in the figure. The noninitial high-norm tokens are labeled in red. Results of additional models is presented in Appendix F.

*Figure 8.* Norm of output tokens of FFN at layer-2 of LLaMA2-7B using the right singular vectors of $F$ as input tokens to FFN. More examples are provided in Appendix G.

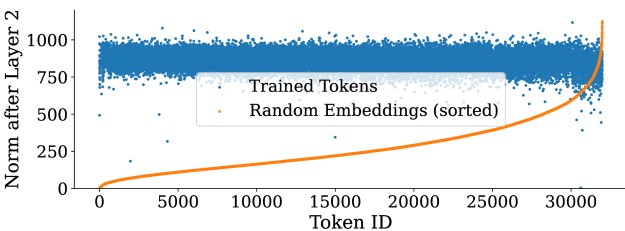

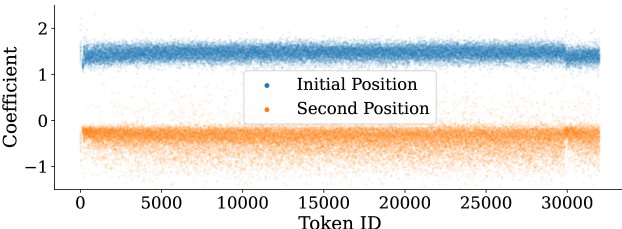

*Figure 7.* Attention-related exploding path of LLaMA2-7B for initial tokens. The $y$-axis is the norm of each token at the output of layer-2. We also show the norm (sorted) of 32,000 random input token embeddings (not learned by the network) after layer-2. See more figures in Appendix F.

*Figure 9.* Coefficient of tokens projected to the leading right singular vector of $F$ just before the FFN in layer-2 of LLaMA2-7B. Analysis for additional models is provided in Appendix G.

### 3.3. *Development*: The Two Types of Exploding Paths

We now focus on the development of the high-norm tokens before the "explosion". To this end, we refer to the computation path related to the appearance of high norms between the input token embedding and the explosion layer as the *exploding path*. We differentiate two types of high-norm tokens: the *initial* token and the *noninitial* high-norm tokens. An example of the noninitial high-norm token is the delimiter token '.' in LLaMA2-7B. We observed that the first occurrence of the '.' token has a high norm in the intermediate hidden states, regardless of the content before it. This implies that *the self-attention layers, the only places where inter-token interactions occur, play an insignificant role in noninitial high-norm tokens*.

Consider the decomposition $L = I + R$ in Eq. (3), and let $x$ be a high-norm token input to a decay layer. If after the layer, the token does not have a high norm anymore, then $Lx \approx 0$, and therefore $Rx \approx -x$. This implies that *the high-norm token $x$ should be an eigenvector of the residual matrix $R$ with a negative eigenvalue*.

To verify this intuition, we compute the minimum angle between the eigenvectors of the residual matrix $R$ and the empirical high-norm direction for LLaMA2-7B. In Fig. 5, we plot these angles for different layers, together with the corresponding eigenvalues. At the decay layer-31, the eigenvector of the residual matrix $R$ forms a small angle with the empirical high-norm direction, with a relatively large negative eigenvalue[2]. We verify this for other models in Appendix E. A corollary of this observation is that, *if the input to the decay layer is not a high-norm token, it will not produce a new high-norm token*, *i.e.*, the layer specializes in shrinking the token norm in the high-norm direction.

To verify this argument, we remove the self-attention layers and feed each token available in the tokenizer individually to the model. Without self-attention layers, all tokens in a sequence effectively behave as independent single-token within the network. If the noninitial high-norm tokens are indeed unrelated to self-attentions, then we expect that the noninitial high-norm tokens retain their high norms after removing self-attention layers from the model. This is verified in Fig. 6, where we plot the norm of the hidden states after the explosion layer for all possible input tokens. Out of the five tokens with the highest norm, four of them belong to noninitial high-norm tokens, including '.', '。', ''

---

[2]For simplicity, we only consider the real part of eigenvalues and eigenvectors, having observed that the imaginary part of the eigenvalues and eigenvectors of interest is close to zero.

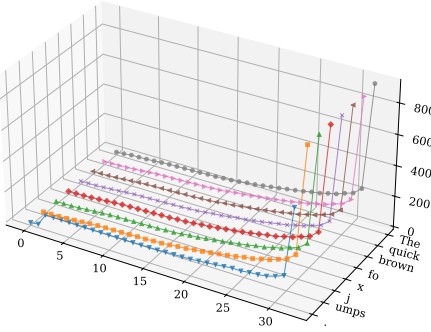

*Figure 10.* Norm of tokens after removing the component on the explosion subspace before layer-2's FFN in LLaMA2-7B. High-norm tokens in the intermediate layers disappear, however, the scale of the final outputs becomes abnormal, leading to low-quality generated text.

and '\n'. Among them, the Chinese delimiter token '。' and the special token '' are newly discovered high-norm tokens that were not identified in previous work.

Given that the high norm of noninitial tokens is unrelated to self-attention, such tokens can also be analyzed under the single-token assumption. Therefore, our methodology for predicting the high-norm directions can be applied to both the initial token and the noninitial high-norm tokens. This sheds light on why the two types of high-norm tokens share the same direction.

For the initial token, we observed the exploding path to be related to self-attention. For example, in Fig. 7, we plot the norm of all the trained tokens after the explosion layer-2 in LLaMA2-7B. Almost all tokens have a high norm. This aligns with the observation that the initial token always has a high norm, regardless of the word it encodes. Comparing Fig. 7 with Fig. 6, we can see that the initial high-norm tokens lose their high norms (except for a few noninitial high-norm tokens) when the self-attention layers are removed from the model. This indicates that self-attention is indispensable for initial tokens to have high norms.

Surprisingly, even a random token embedding that is not learned by the network also has a higher norm than a noninitial normal token, whose average norm is $46.88$ in the explosion layer-2 of LLaMA2-7B, although the norm of the random token may be lower than that of the learned tokens. The analysis for other models is provided in Appendix F.

### 3.4. *Trigger*: **The Explosion Subspace**

Let us now focus on the explosion layer where the norm of tokens undergoes a sharp increase. Tracking the change in norm within that transformer layer, we observed that the abrupt increase occurs in the FFN module. For example, in the explosion layer of LLaMA2-7B (layer 2), the average

norm of the initial tokens before/after FFN is $3.49/932.25$, respectively. Let $F$ be the linear approximation of this FFN module following Eq. (2), and the SVD of $F$ be $F = U\Sigma V^T$. The columns of $V$ form an orthonormal basis of the input space. We feed these unit-length base vectors to the FFN and plot their output norms in Fig. 8. Note that the leading right singular vector of $F$ is the only direction that undergoes norm explosion when passed through FFN. We thus refer to the 1-dimensional subspace spanned by this vector as the *explosion subspace*. This lets us conjecture that *the high-norm tokens have a large component in the explosion subspace, while the normal tokens do not*. This is verified in Fig. 9. For any token at the initial position in the input text (which is thus a high-norm token), the projection of its feature just before the FFN module at layer-2 onto the explosion subspace has a large coefficient (around 1.5). However, when the same token is placed at the second position in the input (where it is not a high-norm token, except for the few noninitial high-norm tokens discussed in Section 3.3), the coefficient is much smaller.

One potential way to avoid high-norm in intermediate layers could therefore be to remove the component in the explosion subspace just before the FFN at the explosion layer. An example of the resulting token norms is shown in Fig. 10, where the high norms in the intermediate layers indeed disappear. However, we observe that the network loses its ability to generate high-quality text and instead produces random texts. This shows that *high-norm tokens are important for the performance of an existing trained model*. This observation echoes the experiments done by Sun et al. (2024), where the authors set the massive activations to zero and find performance degradations.

In summary, the life cycle of high-norm tokens can be outlined as follows. Firstly, the potential high-norm tokens, either the tokens at the initial position of the input sequence, or those noninitial high-norm tokens, develop a sufficiently large component in the explosion subspace. Then, this component explodes, yielding a high-norm token, whose direction can be predicted by the singular defect theory. Finally, the decay layer produces a negative direction to neutralize the high-norm token, which is described by the negative eigenvalue and eigenvector of that layer.

## 4. Discussion

**High-Norm Tokens During Training** High-norm tokens in ViTs appear only in the later stage of training and only in large model variants (Darcet et al., 2024). However, *high-norm tokens in LLMs emerge early in training and across a variety of model sizes*, as demonstrated in Fig. 11 using the suite of Pythia models (Biderman et al., 2023). After 5K out of a total of 143K iterations, the norms of both the initial token and the token '.' have become significantly

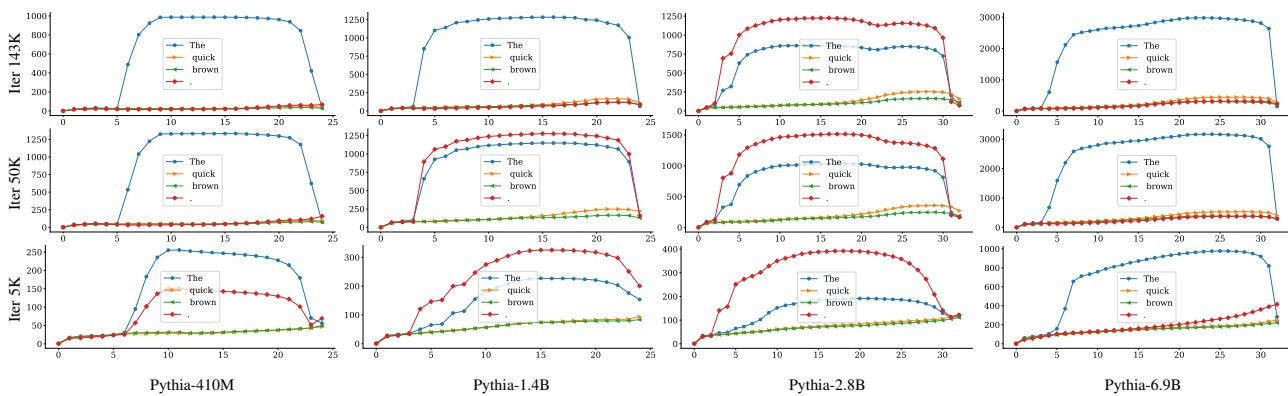

*Figure 11.* High norm tokens in the Pythia model suite. Each row shows a different training iteration, with the top row being the final state at 143K iteration. Each column is a different model size, including 410M, 1.4B, 2.8B, and 6.9B.

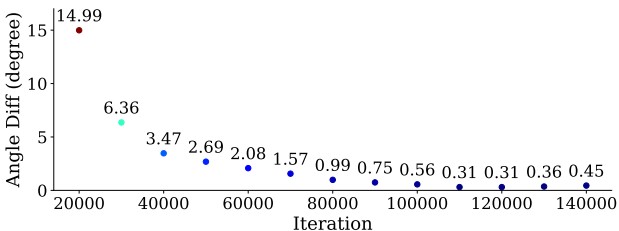

*Figure 12.* Direction of high-norm tokens stabilizes during training. The $y$-axis is the angle between the empirical high-norm directions of adjacent Pythia-1B checkpoints at a $10,000$ interval.

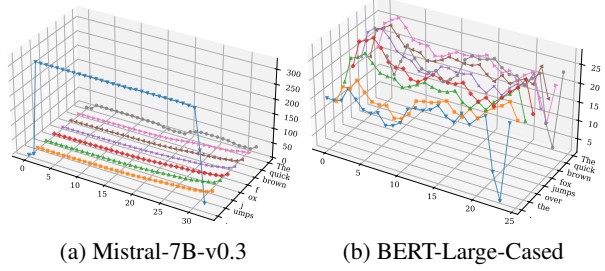

(a) Mistral-7B-v0.3      (b) BERT-Large-Cased

*Figure 13.* (a) Modifying the causal self-attention mechanism affects the emergence of high-norm tokens. (b) Bi-directional models do not show high-norm tokens. More examples are in Appendix B.

higher than that of the other tokens in the hidden states. Furthermore, analyzing the change in the high-norm tokens as training progresses, we found that *the high-norm direction gradually stabilizes during training*, as shown in Fig. 12.

**Which Factor Matters for the Emergence of High-Norm?** From our extensive experiments with various LLMs, we observed that the presence of high-norm tokens is not significantly affected by factors such as the training dataset, positional embedding scheme, FFN design, the use of grouped-query attention, parallel attention-FFN structures (Wang & Komatsuzaki, 2022), supervised fine-tuning(SFT), reinforce-

ment learning with human feedback (RLHF), languages, context length, model size, or training iterations. The root cause therefore remains an *open question*. Nevertheless, empirical evidence suggests that *the causal self-attention mechanism may be one of the defining factors for the emergence of high-norm tokens*. Firstly, note that all the high-norm examples we presented so far were trained with causal attention. Secondly, Mistral (Jiang et al., 2023), which uses a sliding window attention mechanism, does not exhibit high-norm tokens among the initial tokens, as shown in Fig. 13. This indicates that modifying the causal self-attention mechanism affects the emergence pattern of high-norm tokens. Thirdly, we did not observe high-norm tokens in bidirectional models, such as BERT, DistilBERT (Sanh et al., 2019), and RoBERTa (Liu, 2019). This implies that masked language modeling does not induce high-norm tokens as in causal language modeling.

## 5. Applications

In this section, we showcase two applications motivated by the insights of previous sections. In Section 5.1, we improve quantization by specializing it according to the explosion and decay layers. In Section 5.2, we show that the direction of high-norm tokens can be used as a signature of an LLM.

### 5.1. Improving the Design of Quantization

LLMs are memory and computation intensive because of their huge number of parameters. Quantization is a common technique to reduce memory footprint and improve inference speed. Quantization in LLM mainly follows two settings: W8A8 quantization (Dettmers et al., 2022; Xiao et al., 2023), where both the weights and activations are quantized to 8-bit; and low-bit weight-only quantization (Lin et al., 2024; Frantar et al., 2023). We focus on W8A8 quantization. Converting a model from 16-bit floating-point to 8-bit integer can reduce the model size by half. Int8 quantization (Ja-

cob et al., 2018) can be expressed as $\bar{X}^{\text{int8}} = \left\lfloor \frac{X^{\text{fp16}}}{\Delta} \right\rceil$, where $\Delta = \frac{\max(|X|)}{2^{N-1}-1}$ is the quantization step size. Note that $\Delta$ is determined by the maximum absolute value of $X$. Therefore, outliers in the input tensor will lead to non-negligible underflow and cause large quantization errors.

Outlier channels in quantization are different from high-norm tokens (Sun et al., 2024). High-norm tokens refer to hidden states after each transformer layer, whereas outlier channels denote the input activations to linear layers within a transformer layer. Most current LLMs adopt pre-LayerNorm, where a normalization layer is prepended before the self-attention and FFN modules. Hence, the input activations for the query, key, value projections in attention, and $W_1$, $W_3$ in FFN, are normalized. These inputs are not high-norm tokens, yet they possess outlier channels.

Outliers in activations are more severe than in weights (Xiao et al., 2023). As such, the quantization of activations is more challenging than that of weights. For example, tensor-wise W8A8 quantization, where both weights and activations are quantized to int8, does not work well for LLaMA3 (Grattafiori et al., 2024). To tackle this issue, LLaMA3 resorts to the row-wise quantization, where the quantization step size is computed across rows of activation and weight matrices. However, hardware support for row-wise quantization is limited. For instance, the FBGEMM (FBG) implementation of row-wise quantization only supports recent GPUs such as Nvidia H100 and AMD MI300X. Furthermore, tensor-wise quantization has lower latency compared to row-wise quantization (Xiao et al., 2023). It is therefore desirable to improve the quantization strategy to make tensor-wise quantization effective.

We found that the main obstacle to tensor-wise quantization in LLMs is related to the outlier channels induced by the high-norm tokens. In the explosion layer and the decay layer, the residual modules produce high-norm tokens (Section 3), and thus, in the middle of the computation, the intermediate token norms can be very large. Specifically, inside FFN, the norm of some tokens after $\text{silu}(W_1 x) \odot (W_3 x)$ is extremely high, hence affecting the subsequent down_proj $F_2 x$.

Our solution is straightforward: we simply allow the down_proj layer $F_2$ of the FFN module in explosion and decay layers to operate at a higher precision. The difference between our strategy and the mixed-precision strategy in LLM.int8() (Dettmers et al., 2022) is that, LLM.int8() decomposes a matrix into an 8-bit part and a 16-bit part, while we operate the (very few) full problematic matrices in a higher precision. We compare the performance of ours with the two tensor-wise quantization in Table 1. The perplexities of both the standard RTN (Round-To-Nearest) and the recent SmoothQuant (Xiao et al., 2023) are significantly reduced after applying our solution.

*Table 1.* Improving tensor-wise quantization by skipping the down_proj matrix $F_2$ of the FFN module in the explosion and decay layers. PPL is the perplexity on the validation set of WikiText2-v1.

| Model | Method | Skip $F_2$ in Layers | PPL↓ |
|-------|--------|---------------------|------|
| | - | - | 5.47 |
| | RTN | - | 10.18 |
| LLaMA2-7B | RTN | (2, 31) | 6.51 |
| | SmoothQuant | - | 13.87 |
| | SmoothQuant | (2, 31) | 6.78 |
| | - | - | 6.14 |
| | RTN | - | 59.38 |
| LLaMA3-8B | RTN | (2, 32) | 8.80 |
| | SmoothQuant | - | 54.99 |
| | SmoothQuant | (2, 32) | 9.14 |

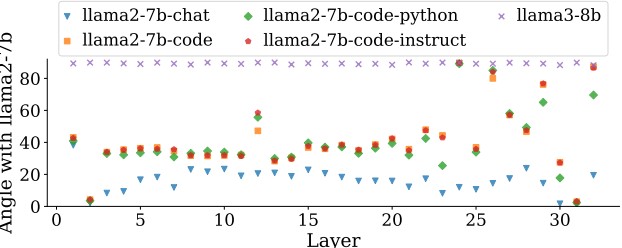

*Figure 14.* The singular defect directions (at explosion layer 2 and decay layer 31 for LLaMA2-7B) remain stable after fine-tuning.

## 5.2. Signature of an LLM

In Section 4, we have shown that the direction of the high-norm tokens gradually stabilizes during training. By extension, this suggests that *the singular defect direction is also robust to fine-tuning*, since fine-tuning is a continuation of the training process. To verify this claim, in Fig. 14, we compare the pairwise angles of the layer-wise singular defect directions between the base model LLaMA2-7B and its fine-tuned variants LLaMA2-7B-Chat, LLaMA2-7B-Code, LLaMA2-7B-Code-Python, LLaMA2-7B-Code-Instruct. For the explosion layer 2 and the decay layer 31, the pairwise angles are very small, which confirms that the direction of the high-norm tokens is stable w.r.t. fine-tuning. Additionally, we added LLaMA3-8B into the comparison, and the pairwise angles are all nearly 90 degrees, suggesting that the singular defect direction could distinguish different models.

An application of this stability is to use the singular defect direction as a signature of an LLM. Specifically, we define the *signature of a LLM* as the empirical high-norm direction. Based on our analysis, the leading left singular vector of either the explosion layer or the decay layer serves as a reliable approximation of the empirical high-norm direction. This approach offers two key advantages: it is data-independent and eliminates the need for manually selecting a threshold

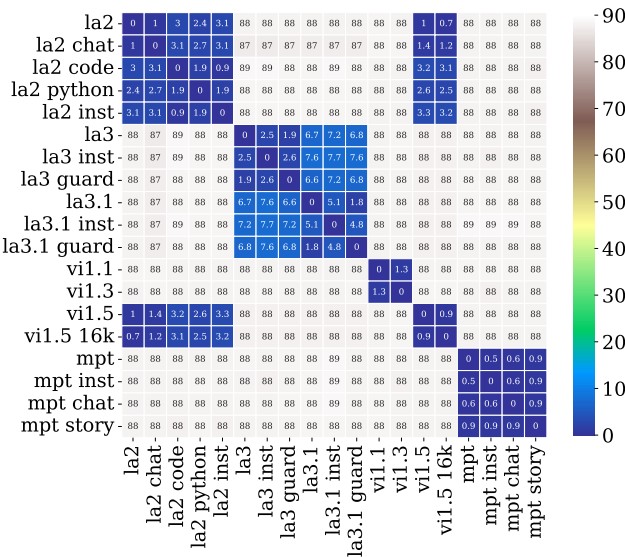

*Figure 15.* Model distances based on angles between model signatures. la2, la3, la3.1, vi1.1, vi1.3, vi1.5, mpt mean LLaMA2-7B, LLaMA3-8B, LLaMA3.1-8B, Vicuna-7B, MPT-7B, respectively. We can see that the distances within each model series are very small, as they originate from the same base models.

to define high norms. By defining the distance between two models (with the same hidden dimension) as the acute angle between their signature vectors, we can *test whether one model is derived from another by computing their signature distances*. In practice, to avoid having to manually locate the explosion/decay layers, we can simply compute the acute angle between the corresponding layer-wise singular defect directions for all layers and take the minimum value as the distance between the two models.

Fig. 15 presents the pairwise distances among several series of LLMs, including the LLaMA2-7B, LLaMA3/3.1-8B, Vicuna-7B, and MPT-7B families. These models naturally cluster into several groups, with models from the same family appearing close to each other. We can infer that LLaMA3 was trained from scratch without reusing the LLaMA2 checkpoint, while LLaMA3.1 was obtained by continuing training from the LLaMA3 checkpoint. Besides, we confirm that Vicuna1.5 is fine-tuned from LLaMA2 instead of Vicuna1.1/1.3. These findings are aligned with the publicly available training details. In addition, we also test distances between two models (Pythia-410M seed1 and seed2) trained with the same data and same architecture, where only the random seeds are different. Their distances are approximately 90 degrees, showcasing the uniqueness of model signatures for independently trained models. Altogether, these results indicate that our LLM signature effectively captures and differentiates the relationships between models, validating its reliability for tracing model lineage.

## 6. Conclusion

We have studied the behavior of high-norm tokens in LLMs from four perspectives, covering their whole life cycle. To this end, we have extended the theory of singular defects from ViTs to LLMs. Specifically, our theory gives a mathematical characterization of the behavior of high-norm tokens that is unique to LLMs. We have extensively verified our theory on a variety of recent models, and our insights have also motivated two practical applications, including a high-norm aware quantization scheme and the design of an LLM signature that may be used to detect model infringement. We hope that our study will motivate advances in the understanding of the internal mechanisms of LLMs.

## Acknowledgements

This work was supported in part by the Swiss National Science Foundation via the grant 200020_214878.

## Impact Statement

This paper presents work whose goal is to advance the field of Machine Learning, in particular, the understanding of LLMs. There are many potential societal consequences of our work, none which we feel must be specifically highlighted here.

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

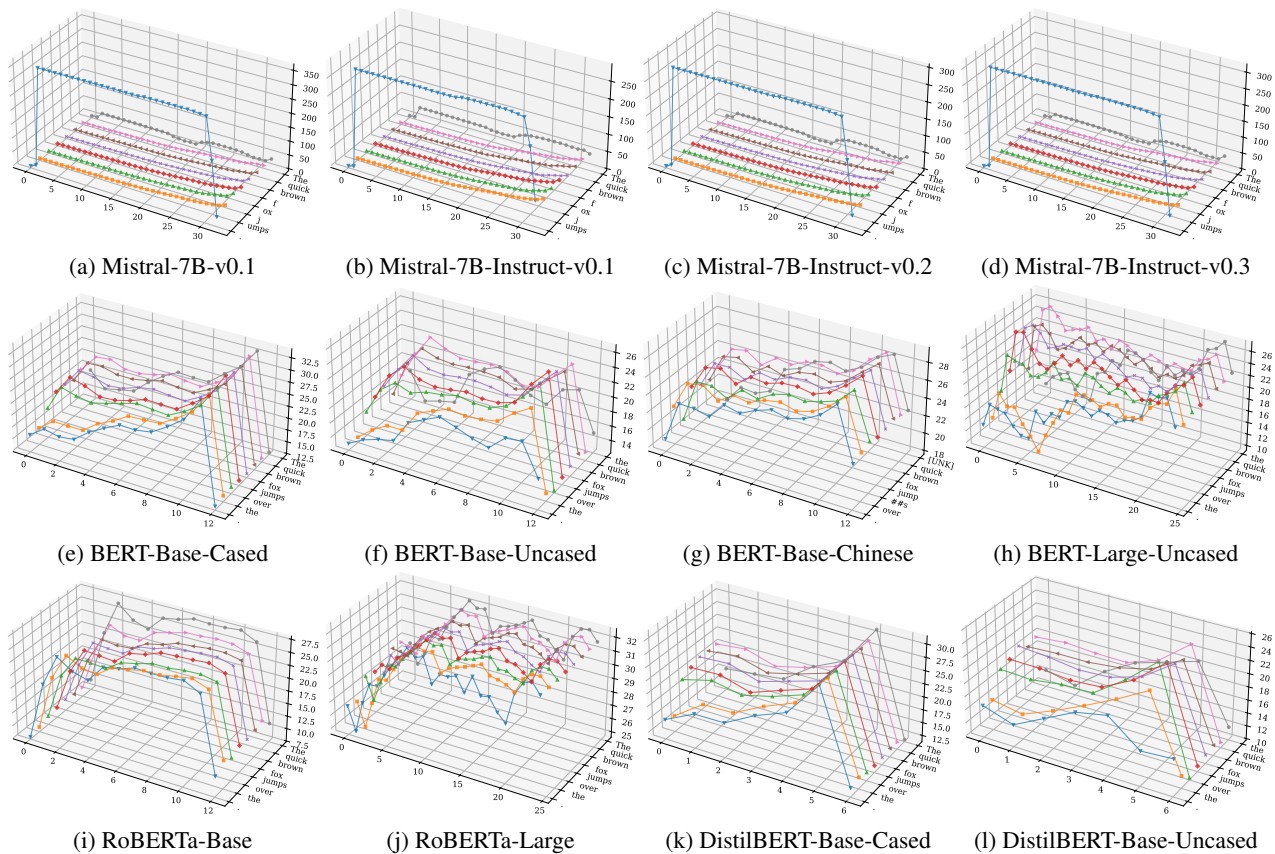

*Figure 16.* (Continuation of Fig. 13). The example token norms in Mistral, BERT, RoBERTa and DistilBERT models.

## A. More Examples of High Norm Tokens

Fig. 17 shows more examples of high-norm tokens in the LLM series. Specifically, it displays a plethora of model variations in the LLaMA2, LLaMA3, LLaMA3.1, LLaMA3.2, Phi3, and Qwen2 series. Without exception, they all exhibit the high-norm phenomenon, especially in the initial token. The patterns of the high-norm look similar if one model is fine-tuned from another, which echos our findings about the stability of high-norm tokens in Section 5.2.

## B. More Examples of Token Norms in Mistral and BERT

Fig. 16 shows the norms of the first few tokens and the token '.' in the sentence 'The quick brown fox jumps over the lazy dog.' for more models in the Mistral, BERT, RoBERTa, and DistilBERT families.

## C. More High-Norm Direction Statistics

Table 2 shows the average pairwise angles between all high-norm tokens for more LLMs. The thresholds are determined by inspecting the visualization of the token norms as in Fig. 1. The results confirm that all the high-norm tokens in one model have very similar directions across all layers and all tokens.

## D. More Examples of Explosion Direction Prediction

Fig. 18 shows the angles between the predicted layer-wise singular defect directions and the empirical high-norm direction for more models. Notice that in some models, such as Phi3-Medium, there are multiple explosion layers and decrease layers, and the angles between the predicted directions and the empirical high-norm directions are close in those layers.

*Table 2.* Average pairwise angles between all high-norm tokens for each LLM. High norm tokens are collected across all layers, and all tokens from 1000 rows in the WikiText2-v1 dataset.

| Model | Threshold | Mean Pairwise Angle (degree) |
|---|---|---|
| LLaMA2-7B | 500 | 3.12 |
| Phi3-Medium | 2500 | 5.31 |
| MPT-7B | 1500 | 2.81 |
| Pythia-160M | 200 | 8.01 |
| Vicuna1.5-7B | 400 | 4.15 |
| Falcon2-11B | 3000 | 4.11 |
| GPT2-Medium | 3000 | 1.49 |
| Qwen2.5-1.5B | 8000 | 1.00 |

## E. More Examples of Eigenvalues in Decay Layer

Fig. 19 shows the minimum angle between the eigenvectors of the linear approximation of the layer residual and the empirical high-norm direction for more models. Their corresponding eigenvalues are also plotted.

## F. More Examples of Exploding Path

Fig. 20 shows the norm of single tokens after the explosion layer when removing all self-attention blocks from the model. The tokens with highest norms are candidates for the noninitial high-norm tokens. Fig. 21 shows the norm of the initial token after the explosion layer. All trained tokens are plotted. Besides, a number of random input embeddings are also used as initial tokens, and their norms after the explosion layer are plotted.

## G. More Examples of Explosion Subspace

Fig. 22 shows that the leading right singular vector of the FFN module in the explosion layer ignites the explosion of the token norms for more models. Fig. 23 shows the coefficients of tokens projected to the explosion subspace just before the FFN in the explosion layer of LLMs for more models.

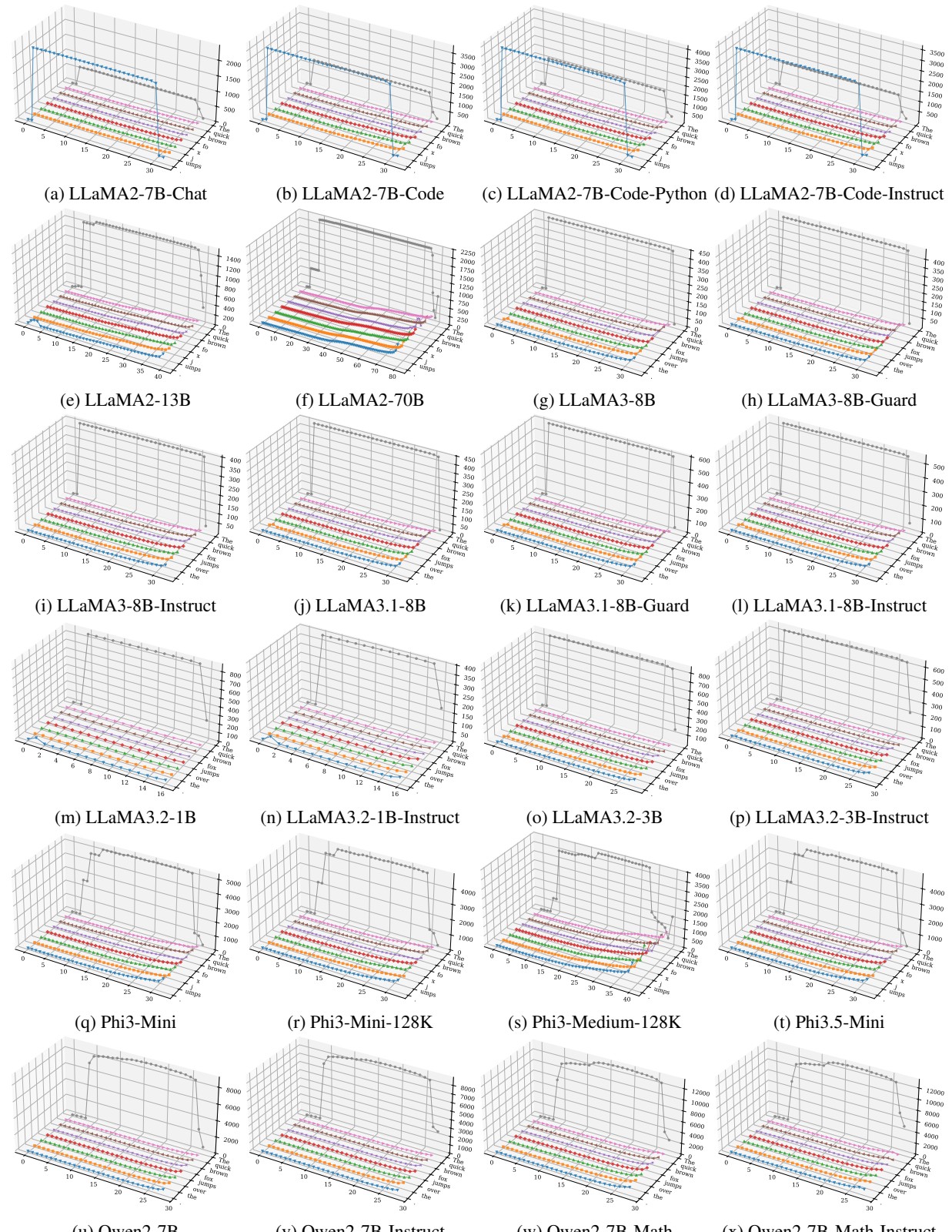

*Figure 17.* (Continuation of Fig. 1). High norm tokens in various LLM series. Each subfigure plots the norm of the first few tokens and the token '.' in the sentence 'The quick brown fox jumps over the lazy dog.'. Here, the token 'The' is the initial token, and it has a high norm in all models. The $x$-axis is the layer id, the $y$-axis shows different tokens, and the $z$-axis is the norm. Layer 0 is the input embedding layer, and the others are transformer layers.

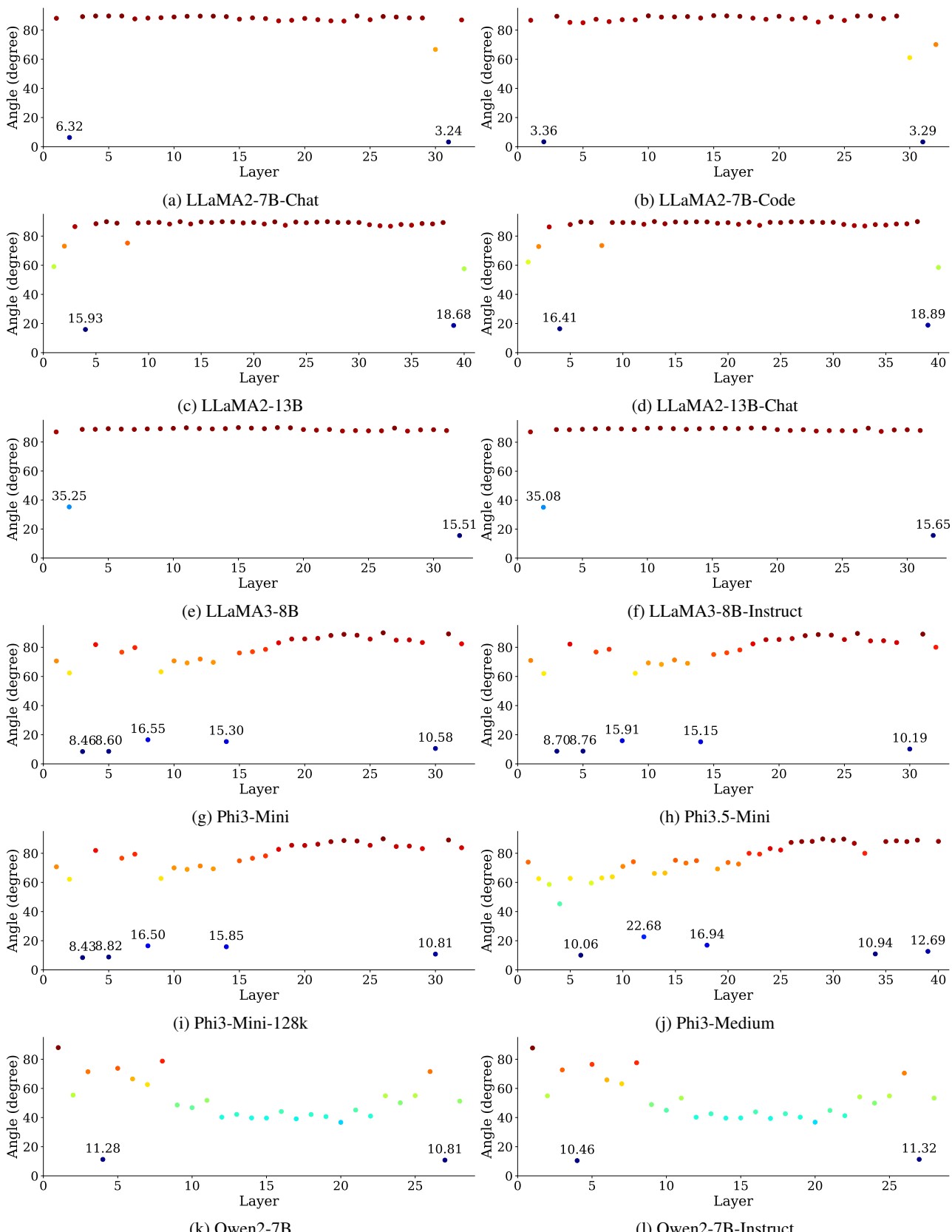

*Figure 18.* (Continuation of Fig. 4). Acute angles between layer-wise singular defect directions and empirical high-norm direction for more models. We can see that small angles align with either the explosion or the decay layer.

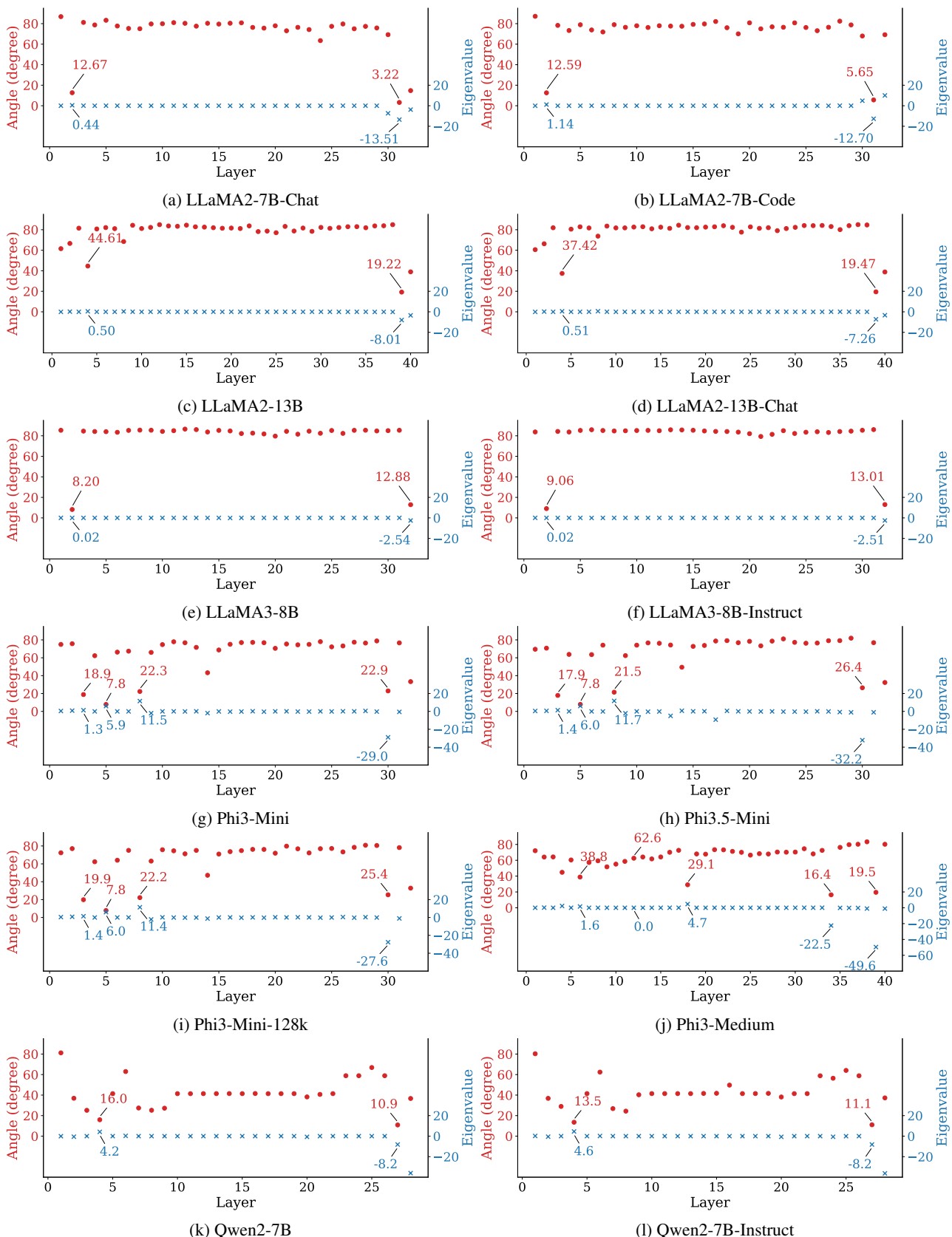

*Figure 19.* (Continuation of Fig. 5). For each layer, the minimum angles between the eigenvectors of $R$ and the empirical high-norm direction are shown in red, and the corresponding eigenvalues are shown in blue. Numbers for the explosion and decay layers are annotated.

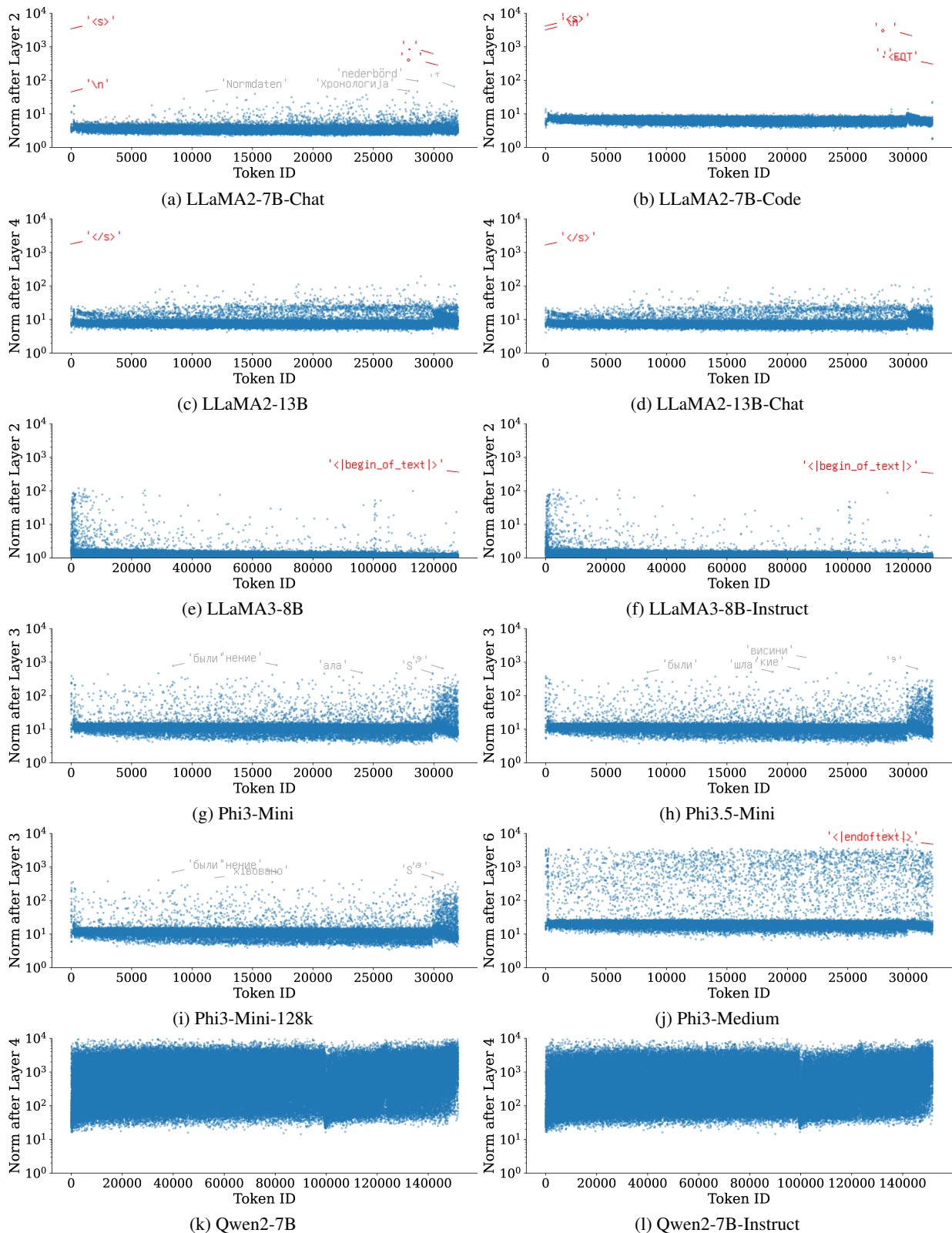

*Figure 20.* (Continuation of Fig. 6). Attention-independent exploding paths. The $y$-axis is the norm of all single tokens after the explosion layer when removing all self-attention blocks from the model. The largest few tokens together with their norms are annotated in the figure. Red color means they are noninitial high-norm tokens.

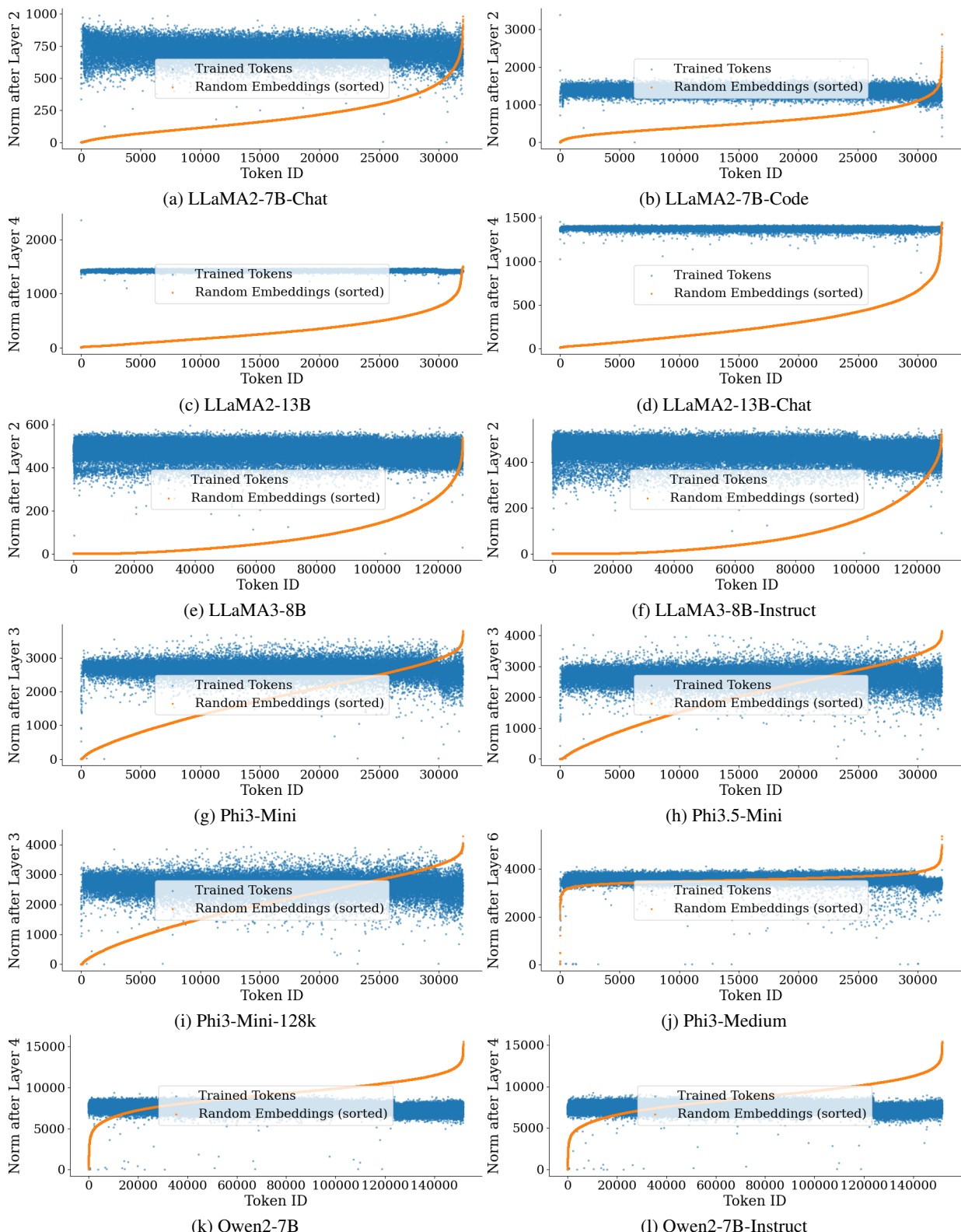

*Figure 21.* (Continuation of Fig. 7). Norm of all trained tokens after the explosion layer when they are used as the initial token in an input sequence. We also plot the norm of a number of random input embeddings (sorted by their output norms) after the explosion layer.

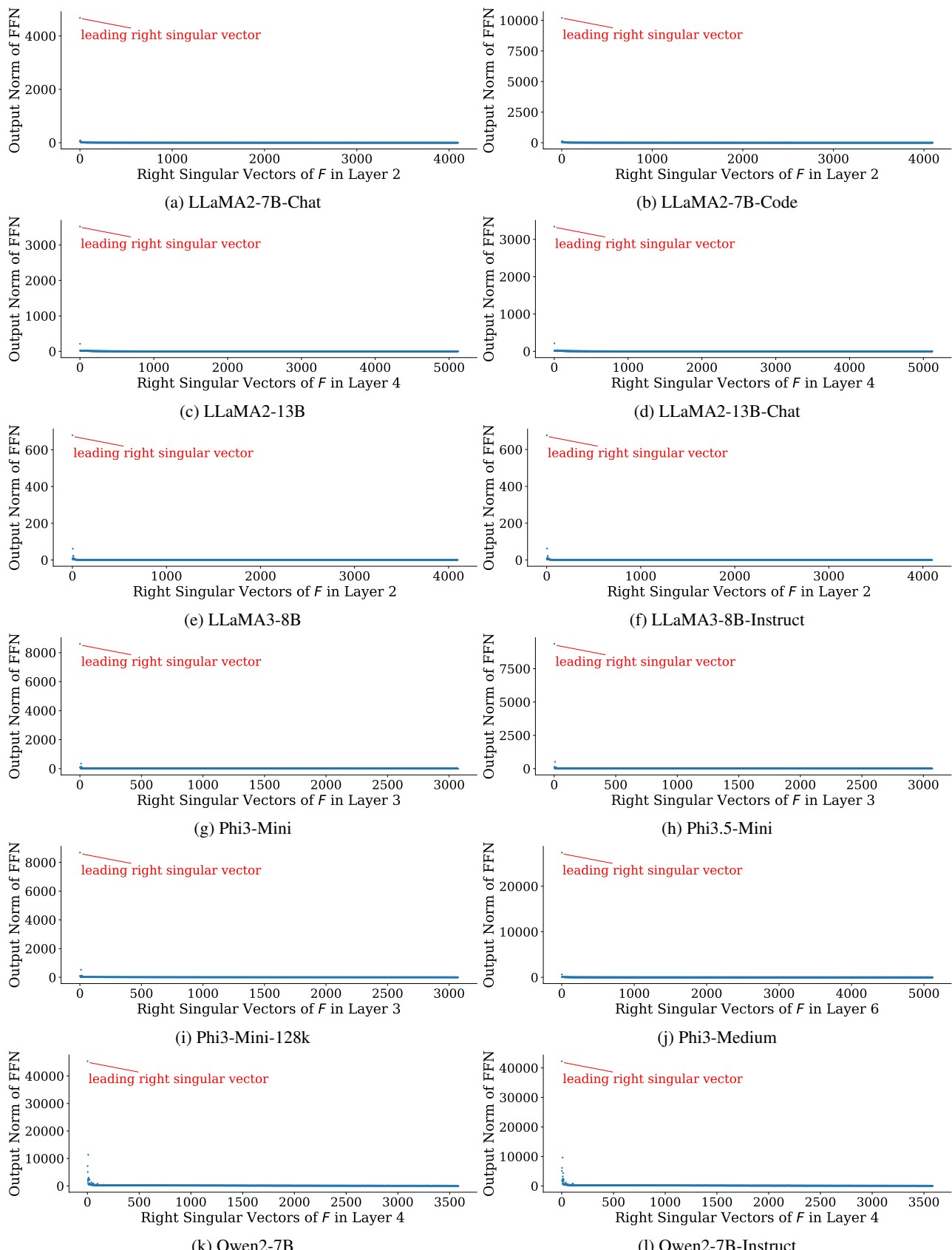

*Figure 22.* (Continuation of Fig. 8). Norm of output tokens of FFN in the explosion layer using the right singular vectors of $F$ as inputs to FFN. We can see that the leading right singular vector of the FFN module in the explosion layer ignites the explosion of the token norms.

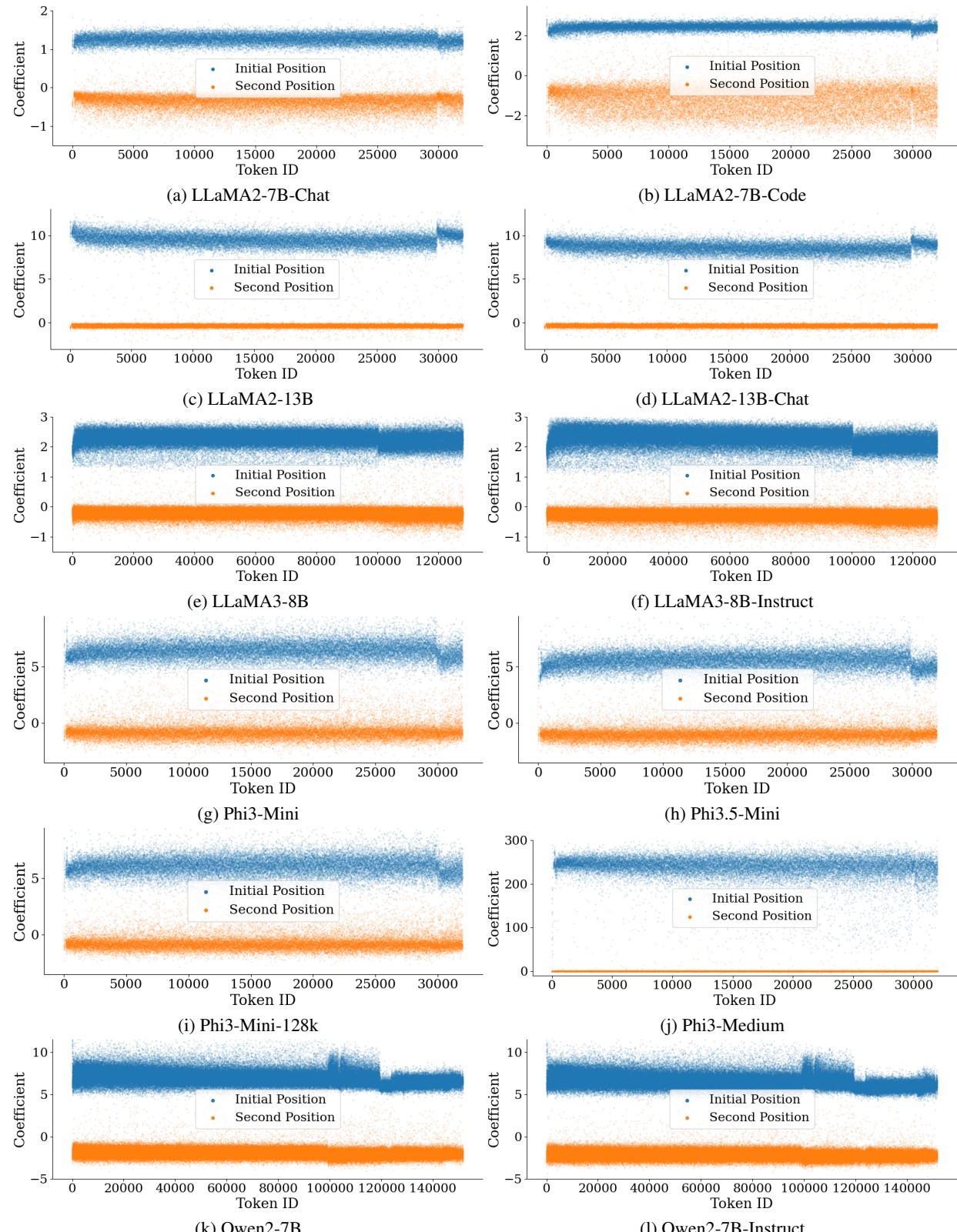

*Figure 23.* (Continuation of Fig. 9). Coefficient of tokens projected to the explosion subspace just before the FFN in the explosion layer of LLMs. The initial tokens have a high component in the explosion subspace.

