# OpenReview forum: "Demystifying Singular Defects in Large Language Models"
_ICML.cc/2025/Conference — ICML 2025 poster_

### Official Review · Reviewer_6zUQ · 2025-02-26

**Overall Recommendation:** 3

**Summary:**

This paper investigates the phenomenon of high-norm tokens in LLMs, identifying key factors that influence their behavior. These factors include singular directions, negative eigenvalues, and distinct computational pathways for initial and non-initial tokens. The study reveals that high-norm tokens are primarily driven by the leading singular vector of specific model components. These insights have practical implications for enhancing quantization schemes and designing LLM signatures.

**Claims And Evidence:**

N/A

**Essential References Not Discussed:**

N/A

**Experimental Designs Or Analyses:**

N/A

**Methods And Evaluation Criteria:**

N/A

**Other Comments Or Suggestions:**

N/A

**Other Strengths And Weaknesses:**

N/A

**Questions For Authors:**

N/A

**Relation To Broader Scientific Literature:**

N/A

**Theoretical Claims:**

N/A

---

> ### Author Rebuttal · Authors · 2025-03-31
>
> Thank you very much for your encouraging comments!

---

### Official Review · Reviewer_NnNK · 2025-03-14

**Overall Recommendation:** 4

**Summary:**

This paper investigates the phenomenon of high-norm tokens in large language models (LLMs), extending the understanding of singular defects from vision transformers (ViTs) to the context of LLMs. Unlike ViTs, where high-norm tokens have been modeled through singular vectors of linear approximations, the causes and characteristics of high-norm tokens in LLMs remain largely unexplored. The authors provide both theoretical insights and empirical validation across various recent models, leading to several key observations:

1. The layer-wise singular direction predicts the abrupt explosion of token norms in LLMs.
2. Negative eigenvalues of a layer explain the sudden decay of high-norm tokens.
3. The computational pathways leading to high-norm tokens differ between initial and non-initial tokens.
4. High-norm tokens are triggered by the right leading singular vector of the matrix approximating the corresponding modules.

The authors demonstrate two such applications: improving quantization schemes and designing LLM signatures. The improved quantization strategy selectively preserves precision for critical layers, enhancing robustness without compromising efficiency. Meanwhile, the stable high-norm directions serve as robust signatures to trace model lineage and detect model infringement.

**Claims And Evidence:**

Yes

**Essential References Not Discussed:**

N/A

**Experimental Designs Or Analyses:**

Yes. No issues are found yet.

**Methods And Evaluation Criteria:**

Yes

**Other Comments Or Suggestions:**

N/A

**Other Strengths And Weaknesses:**

Strengths:

- I think it is a valuable paper that addresses an important yet underexplored topic: the behavior of high-norm tokens in LLMs. The findings provide a lot of new insights and greatly inspires the future works.

Weaknesses:

- This paper hypothesizes that the causal self-attention might be the underlying reason for the emergence of LLM's high-norm tokens. So I just wonder why ViTs (no causal attention) also have the similar high-norm tokens? Does the causal attention really the key factor of this phenomenon?

- I am curious about the high-norm vision tokens in LLaVA-style Large VLMs. The paper shows the difference functions about the high-norm tokens in LLMs and ViTs. While in Large VLMs, some previous works noticed that the vision tokens in VLMs inherit the high-norm tokens greatly from the ViT (the vision encoder). How about the behaviors of these high-norm vision tokens in the language models of VLMs? Are they still working as the vision registers or behaving as the LLM high-norm tokens like claimed in this paper?

- What about the differences between the attention sink[1] and the high-norm tokens discussed in this paper?

- What is the actual role of the the high-norm tokens of LLMs discussed in this paper?

- More discussions are expected, e.g., which kinds of text tokens are easier to trigger the high-norm token phenomenon?

[1] Xiao, et al. Efficient Streaming Language Models with Attention Sinks. ICLR, 2024.

**Questions For Authors:**

See weaknesses.

**Relation To Broader Scientific Literature:**

N/A

**Theoretical Claims:**

Yes. No issues are found yet.

---

> ### Author Rebuttal · Authors · 2025-03-31
>
> Thank you for the insightful comments!
> We will incorporate the suggestions in the revision.
>
> > Is causal attention really the key factor? Why ViTs have similar high-norm tokens?
>
> We hypothesize that causal attention excites high norms in LLM in the following way.
> In the causal formulation, the possible output features at the initial position are finite ($=V$, the vocabulary size, e.g., 32K for LLaMA2).
> The softmax loss increases the norm of these features for each training sample, so their norms grow fast.
> By comparison, the possible output features at the 2nd position are 1 billion ($=V^2$).
> The push towards increasing their norms is thus dispersed, yielding no systematic high norms in the 2nd and subsequent positions.
>
> The high-norm tokens in ViTs could stem from different causes as implied by their drastically different patterns.
> 1. They appear in the late stage of training, while High-norm tokens in LLMs appear in early stage (L324-327).
> 2. They grow progressively layer by layer, while high-norm tokens in LLMs explode abruptly (L157-159).
> 3. They appear at random locations, while high-norm tokens in LLMs mainly locate at the starting position (L160).
> 4. They can be repaired by SINDER without affecting performance, while the presence of high-norm tokens is critical for the performance of LLMs (L310).
> 5. The single-token assumption in ViTs is an ideal assumption to simplify the analysis. By contrast, in LLMs, the single-token assumption is real and valid for the initial token (L157 Column2).
>
> These differences motivate the question "whether the theory of singular defects can be applied to LLMs" (L34-35 Column2) and our study gives an affirmative answer.
>
> > How about the behavior of high-norm vision tokens in the language models of VLMs? Are they working as the vision registers or behaving as the LLM high-norm tokens?
>
> To answer this question, we forwarded an image-text mixed data using LLaVA-v1.5-7b, and located the image patch token with the highest norm in the ViT features.
> Here is its norm (round to integer) at different LLM layers:
>
> ```txt
> 764 (norm of input visual token), 764 (after 1st layer of LLM), 764, 765, 764, 764, 764, 762, 761, 760, 758, 756, 754, 752, 750, 748, 746, 744, 741, 740, 738, 738, 737, 736, 734, 734, 732, 732, 732, 730, 730, 729, 728
> ```
>
> Unlike the high-norm text token in LLM, whose norm explodes and decays, the norm of high-norm vision token stays nearly the same throughout the LLM.
> In addition, the angle between the high-norm vision token and the high-norm text token is kept around 87 degrees across LLM layers.
> Based on these, we hypothesize that high-norm vision tokens neither work as registers nor behave as LLM high-norm tokens.
>
> > What about the differences between the attention sink and the high-norm tokens?
>
> Attention sink refers to the phenomenon that the initial token receives large attention scores.
> It is a side-effect emerging from the fact that the initial tokens have high norms, as discussed in the massive activation paper by Sun et al.
>
> We believe our efforts on demystifying high-norm tokens provide a better understanding of the underlying mechanism of attention sinks.
> For example, in Fig. 12, we show that model trained with window attention (Mistral) does not have initial high-norm tokens.
> As a result, Mistral can process arbitrary context length without the special treatment introduced in the attention sink paper.
>
> > What is the actual role of the high-norm tokens in LLMs?
>
> In Fig. 13, we show that high-norm tokens appear in the very early stages of training.
> We therefore hypothesize that they are a way to accelerate network convergence by injecting some biases.
> To verify this, we added an extra layernorm at the output of the residual branch of each attention block and that of each FFN block to suppress any possible high-norm tokens.
> We trained the modified LLaMA2-1b model from scratch and observed that the loss decreased more slowly compared to the original structure.
>
> > Which kinds of text tokens are easier to trigger the high-norm token phenomenon?
>
> Firstly, as shown in Fig. 7, any token in the initial position will trigger the high-norm phenomenon.
> Secondly, some high-frequency tokens in the corpus such as `\n`, `<s>` may appear as non-initial high-norm tokens.
> Yet, the set of non-initial high-norm tokens may change during training.
> For example, in Fig. 13 Column2, the '.' token is a non-initial high-norm token at training iteration 50k, but is no longer a high-norm token at iteration 143k.
> We will extend Fig. 6 to more LLMs in the revision and below we summarize their non-initial high-norm tokens.
>
> |Model|Non-initial high-norm tokens
> |:-|:-
> |LLaMA2-7B-Chat|`<s>`, `.`, `。`, `\n`
> |LLaMA2-7B-Code|`<s>`, `.`, `。`, `\n`
> |LLaMA2-13B|`</s>`
> |LLaMA2-13B-Chat|`</s>`
> |LLaMA3-8B|`<\|begin_of_text\|>`
> |LLaMA3-8B-Instruct|`<\|begin_of_text\|>`
> |Phi3-Mini|None
> |Phi3.5-Mini|None
> |Phi3-Mini-128k|None
> |Phi3-Medium|`<\|endoftext\|>`
> |Qwen2-7B|None
> |Qwen2-7B-Instruct|None

---

### Official Review · Reviewer_3U5B · 2025-03-15

**Overall Recommendation:** 3

**Summary:**

This paper is a direct follow-up to SINDER (Wang et al, 2024). In this paper, the authors use the tool of “singular defects” to analyze the occurrence of high-norm tokens in language models, which was observed in (Sun et al. 2024). They analyze the weights of the model to understand where high-norm tokens come from, and how they are decayed. They find that certain weight matrix properties can be used to predict which layers produce high norm tokens. Finally, the authors demonstrate that high-norm tokens can be used to (1) improve quantization by not quantizing the layers responsible for handling the high-norm tokens and (2) act as a signature between language models, since it does not change much in fine tuning.

**Claims And Evidence:**

The claims are generally supported by evidence. However, in my view much of the findings are "re-discoveries" of findings that were initially observed in Sun et al. rather than new findings.

In particular, Section 3.1 says that it identifies an "explanation for why there exist a set of fixed channels with high activations" - I don't think this is supported by the content of the paper. The fact that there is an explosion layer and a decay layer already was already identifed in Sun et al (2024) (Page 4, Fig 4). The singular defect analysis explains why tokens with high activations in those particular channels do not change, but it doesn't explain why those particular channels contain high activations, or how those occur in the first place.

Similarly, the finding in Section 3.3 that self attention doesn't affect high norm tokens seems to be a clear corollary of the finding in Sun et al that these tokens act as "biases" and are somewhat input independent (Page 6).

Finally, the notion of the "explosion subspace" (section 3.4) and removing the relevant component is equivalent to the experiment in Sun et al that zeros out the high-norm channels of these tokens and finds a drop in performance; as expected. this finding exactly matches that of Sun et al.

In summary, I think a lot of the claims in this paper are certainly backed by evidence; it's just that these aren't new findings and just confirmation of phenomena already observed in prior work, couched in a different theoretical framework.

**Essential References Not Discussed:**

I think this paper is missing a key reference: LLM.int8(): 8-bit Matrix Multiplication for Transformers at Scale
by Dettmers et al.

This paper identifies the exact phenomeon in the Applications section - that quantization can be improved by acknowledging the existence of large activations, and keeping those in higher precision. As such, the application related to quantization is not new at all.

**Experimental Designs Or Analyses:**

I think the experiments and analysis are similar to those in SINDER, Darcet et al, and Sun et al; these relate to taking a pre-trained model and probing different parts of it. This seems reasonably sound to me.

**Methods And Evaluation Criteria:**

This paper is mostly based on analysis and measuring certain properties and phenomena that occur inside a fixed language model. The evaluation of these seems reasonable.

This paper doesn’t really require strong benchmarking in its experiments except for the applications section. I am not familiar with quantization benchmarks or LLM signature verification, but those experiments seem reasonable to me.

**Other Comments Or Suggestions:**

1. The figures are far too small. These should either be larger or omitted from the paper. Figures 6-9 are also very small; the captions of these figures should not be the same size as the figures themselves.

**Other Strengths And Weaknesses:**

Strengths: The paper does do a careful analysis of the properties of these high norm tokens, and adds valuable analysis. I think the applications section is particularly strong, and it shows that acknowledging the existence of these special tokens is crucial for modifying pre-trained weights.

Weaknesses:
My first main issue is that many of the empirical findings in this work are not new and were already discovered in Sun et al; the paper should be re-written to reflect that these confirm prior findings and are not observations of new phenomena, as I note in the Claims+Evidence section.

My second main issue is that the concept of the paper itself seems rather tenuous. It amounts to looking at the weight matrices and finding that certain values in it lead to large norm tokens, which is not actually the question of interest posed in Sun et al (2024) or Darcet et al (2023). That question is: why does this occur during training? Why do the weight matrices have these properties in the first place?. This  more interesting question remains unanswered (as noted in Section 4) and thus limits the value of this analysis towards further understanding the occurrence of high-norm tokens; the analysis in this paper doesn’t really add much beyond what was introduced in the SINDER paper.

The contribution of this paper instead is showing that you can apply linear algebra tools to analyze the behavior of activations under very strong assumptions (linearizing the attention and FFN blocks). This was already demonstrated in SINDER, and this paper applies it to the FFN layer. I don’t think there is much more theoretical contribution in this work beyond what was originally proposed in SINDER.

Finally, the applications section does not not acknowledge prior work of Dettmers et al (noted in the above essential references section) which already applies mixed precision quantization to handle the existence of particularly high norm features. While this is not exactly the same (high norm channels/tokens vs features), this is essentially the same idea.

As such, I rate the paper as a weak reject.

**Questions For Authors:**

The initial token analysis makes sense, since the linearization model only applies to one token. Doesn’t this make the analysis of non-initial tokens very challenging? I’m very confused as to how we can expect to draw any meaningful conclusions from this.

Isn’t the finding in section 3.1 / FIgure 4 that high norm tokens shared the same direction already stated in Sun et al? They found that these HNs occur in the same locations in the vectors - this isn’t a new finding. The notion of “empirical high norm direction” is already considered in that work too, just under a different name.

 Isn’t the finding that “self attention plays an insignificant role in non-initial HN tokens” (Section 3.3) the same as the finding from Sun et al. that high-norm tokens act as fixed biases? They also found that they are independent of input values.

Isn’t the finding that removing the “explosion subspace component” degrades performance essentially equivalent to the experiment of Sun et al that zeroes out the high-norm tokens?

How does the quantization procedure differ from that of LLM.int8()?

This is more of a comment, but the main thing that would change my evaluation of this paper is a better understanding of how it fits in to the broader literature and what exactly new it contributes. The analysis section as I currently understand it doesn’t demonstrate any meaningful new understanding of why high-norm tokens occur in the first place, and many of the experimental findings mirror those identified in Sun et al. What exactly does this work contribute beyond SINDER and Sun et al besides straightforward application to language models?

**Relation To Broader Scientific Literature:**

This paper is a direct extension / follow-up to SINDER. The original papers that described the phenomenon of large-norm tokens were Vision Transformers Need Registers (2023) and Massive Activations in LLMs (2024). These papers explored the existence and probed the function of these high norm tokens in language models. Notably, these papers did not try to find how these high norm tokens came to be during the training process.

A more recent paper (Wang et al, 2024), SINDER, introduced the terminology of “defective” tokens and analyzed the weights of the revelation model. SINDER attempts to ‘repair’ the model by imposing a strong regularization on the singular values of the weight matrices. This paper applies the same technique, but to language models.

**Theoretical Claims:**

While the “theory of singular defects” is a main tool applied in this paper, it was developed originally in SINDER and does not really require rigorous proof. Although it relies on somewhat strong assumptions (ie single token, heavy linearization) I think it is a reasonable tool to analyze properties of the weights of a model, especially since it clearly identifies which layers are responsible for producing and decaying high norm tokens.

---

> ### Author Rebuttal · Authors · 2025-03-31
>
> Thank you for the detailed and thoughtful comments and we are encouraged by the remark that "applications section is particularly strong".
> We will incorporate the suggestions in the revision.
>
> > How it fits into the broader literature and what exactly new it contributes
>
> While Sun et al. spotted high-norm tokens and SINDER provided a basic tool, our work advances the understanding from three aspects:
>
> * **Structural Characterization**. We introduce explosion subspace and decay eigenvalue analysis, providing a structured framework for high-norm token evolution in LLMs.
> * **Causal Insights**. We conjectured a link between high-norm token emergence and the causal self-attention.
> * **Practical Applications**. We demonstrate their impact on quantization and introduce model signatures for lineage tracing.
>
> We highlight our unique contributions and whether they relate to SINDER and Sun et al. below.
>
> |Type|Source|Our Contribution|Sun et al.|SINDER|Comment
> |:-|:-|:-|:-|:-|:-
> |analysis|Sec. 3.1|The layer-wise singular direction predicts the explosion direction|x|related|SINDER studied ViT. We extend it to LLMs
> ||Sec. 3.2|Negative eigenvalues of a layer explain its sudden decay|x|x
> ||Sec. 3.3|The explosion pathway of the initial token links to self-attention, whereas that of the noninitial high-norm token is unrelated to self-attention|x|x
> ||Sec. 3.4|High-norm tokens are triggered by the explosion subspace|x|x
> |observation|L326|High-norm tokens emerge early in training|x|x
> ||L303 Column2, L379 Column2|The high-norm direction stabilizes during training and is robust to fine-tuning|x|x
> ||L267-274|A systematic analysis that reveals all noninitial high-norm tokens|partial|x|Sun et al. failed to find tokens `。` and `<s>` for LLaMA2-7b
> |application|Sec. 5.1|Improvement on quantization schemes|x|x
> ||Sec. 5.2|Design of LLM signatures|x|x
>
> > The linear model applies to one token and makes the analysis of noninitial tokens challenging
>
> Fig. 6 shows that the noninitial high-norm tokens retain their high norms after removing all self-attentions.
> They effectively behave as independent single-token within the network (L262-264), which makes their analysis by linear model feasible.
>
> > Sun et al. found that HNs occur in the same locations in the vectors, which is the same as "high norm tokens share the same direction"
>
> Their observation is **not sufficient** to reach our conclusion that "high norm tokens share the same direction".
> Firstly, they show no evidence that the relative magnitude of massive locations is consistent.
> For example, Tab. 1 and 2 of Sun et al. track the scalar magnitude by sorting, which discards the channel locations.
> Secondly, they did not quantify the influence of the non-massive locations (which are in majority) on vector direction.
> As such, they cannot rigorously infer our conclusion.
>
> > "Self-attention plays an insignificant role in noninitial HN tokens" is the same as Sun et al's "high-norm tokens act as fixed biases"
>
> We respectfully disagree.
> Our statement refers to the development of noninitial high-norm tokens, namely, they still exhibit high norm after removing self-attention (Fig. 6).
> Contrastingly, self-attention is **indispensable** for initial high-norm tokens: When it is removed, initial tokens lose high norms (Figs. 7 and 6).
> This underscores a fundamental difference to Sun et al's.
>
> > The findings from removing the explosion subspace are equivalent to Sun et al.'s zeroing out massive activations
>
> We will add the following clarification in the revision. "This observation echoes the experiments done by Sun et al., where the authors set the massive activations to zero".
>
> > How does the quantization procedure differ from that of LLM.int8()?
>
> The core innovation of LLM.int8() is to decompose a matrix into an 8-bit part and a 16-bit part.
> We did not use this trick.
> Besides, they rely on row-wise quantization whereas we use tensor-wise quantization (L354-357 Column2).
> Finally, more than 50% of their matrices are affected (Fig. 3 in their paper), whereas ours only affects 2 matrices.
> We will add LLM.int8() to the related work.
>
> > Why high-norm tokens occur during training in the first place?
>
> We have ruled out 11 factors (L308-313 Column2) and singled out the most probable reason: The causal self-attention.
> It is evidenced by the impact of different attention formulations (L318-329).
> With causal attention, the possible output features for the initial token are finite ($=V$, the vocabulary size, e.g., 32K for LLaMA2).
> The softmax loss pushes the norm of these features to grow for each training sample, resulting in high-norms.
> By comparison, the count of possible output at the 2nd position is 1 billion ($=V^2$), which disperses the push toward increasing their norms.
>
> > The "explanation for why there exists a set of fixed channels with high activations" is not supported
>
> To avoid confusion, we will revise it to "This explains why the set of high-activation channels observed in Sun et al. is *fixed*".

---

> > ### Comment · Reviewer_3U5B · 2025-04-05
> >
> > The rebuttal mostly resolved my concerns, and I will raise my score to Weak Accept.
> >
> > I think that the paper should still be revised to very clearly delinate the differences between prior work and this paper, which adds an incremental contribution to existing works. Furthermore, the applications section should be seriously re-framed. Currently, it's written as if this paper is the first to observe that high-norm tokens matter for quantizations, which is simply not true. The contribution of the applications section should be framed as incremental improvement on top of existing work.

---

> > > ### Author Response · Authors · 2025-04-06
> > >
> > > We are deeply encouraged that our rebuttal has resolved most of your concerns and we sincerely appreciate your decision to raise the score to 3. In the revision, we will very clearly differentiate our contribution from previous works. Furthermore, we will expand the discussion to related works on the quantization application and properly clarify its improvement on top of the existing works. Thank you again for your valuable time and efforts in reviewing our paper.

---

### Decision · Program_Chairs · 2025-05-01

**Decision:**

Accept (poster)

**Comment:**

All reviewers agreed this paper should be accepted: it addresses an important problem, the analysis is carefully done, and the applications section is strong. The main concerns were around framing: reviewer 3U5B thought that the novelty claims of the paper were a bit exaggerated, and the authors clarified their contributions beyond prior work.